# Nonionizing Electromagnetic Field: A Promising Alternative for Growing Control Yeast

**DOI:** 10.3390/jof7040281

**Published:** 2021-04-08

**Authors:** Byron Riffo, Consuelo Henríquez, Renato Chávez, Rubén Peña, Marcela Sangorrín, Carlos Gil-Duran, Arturo Rodríguez, María Angélica Ganga

**Affiliations:** 1Departamento de Ciencia y Tecnología de los Alimentos, Facultad Tecnológica, Universidad de Santiago de Chile, Alameda 3363, Santiago 9170022, Chile; byron.riffo@usach.cl (B.R.); consuelo.henriquez@usach.cl (C.H.); ruben.pena@usach.cl (R.P.); 2Departamento de Biología, Facultad de Química y Biología, Universidad de Santiago de Chile, Alameda 3363, Santiago 9170022, Chile; renato.chavez@usach.cl (R.C.); carlos.gil.d@usach.cl (C.G.-D.); 3Instituto de Investigación y Desarrollo en Ingeniería de Procesos, Biotecnología y Energías Alternativas PROBIEN, Universidad Nacional del Comahue, Consejo Nacional de Investigaciones Científicas y Tecnológicas, Buenos Aires 1400, Argentina; marcela.sangorrin@probien.gob.ar; 4Departamento de Tecnologías Industriales, Facultad Tecnológica, Universidad de Santiago de Chile, Alameda 3363, Santiago 9170022, Chile

**Keywords:** electromagnetic fields, food quality, membrane disruption, radiofrequency waves, *Saccharomyces cerevisiae*, spoilage control

## Abstract

In the food industry, some fungi are considered to be common spoilage microorganisms which reduce the shelf life of products. To avoid this outcome, different technologies are being developed to control their growth. Electromagnetic fields (EMF) have been used to combat bacterial growth, but there are few studies on yeasts and their possible action mechanisms. For this reason, we studied the effect of EMF between 1 to 5.9 GHz bands on the growth of *Saccharomyces cerevisiae* yeast and observed that all the frequencies of the band used cause the reduction of the viability of this yeast. In addition, we observed that the distance between the antenna and the sample is an important factor to consider to control the growing yeast. By using transmission electron microscopy, we found that the EMF caused a loss of continuity of the yeast cell membrane. Therefore, EMF may be used as a control method for yeast growth.

## 1. Introduction

A worldwide trend in food consumption is oriented towards developing nutrient and healthy products with the least possible necessary technological intervention, and this trend has been adapted easily to the way of life in the modern world [1]. This scenario has established new production, distribution, and quality control outcomes where they all have a very important cross-sectional element, which is the deterioration of the products in every process. This deterioration is closely related to the growth of opportunistic microorganisms that cause physical, chemical, and/or sanitary alterations [2,3,4,5]. Yeasts, which are among the spoilage microorganisms commonly found in foods, have an advantage in their diversity and number, reaching values between 10^3^ and 10^5^ CFU/g [5,6]. Among the genera, *Saccharomyces*, *Candida*, *Debaryomyces*, *Dekkera*, *Hanseniaspora*, *Kluyveromyces*, *Meyerozyma*, *Pichia*, *Rhodotorula*, *Torulaspora,* and *Zygosaccharomyces* can be named, but *Saccharomyces cerevisiae* species as the predominant microorganism in general [5,6,7]. These microorganisms are mostly responsible for organoleptic deterioration because of their ability to produce gases, acids, turbidity, browning, and various organic compounds that give a “fermented flavor” [5]. 

Various technologies have been implemented to avoid the proliferation of food contaminating microorganisms, with pasteurization as the best-known process [2]. Therefore, strategies like treatment with ozone, gaseous chlorine dioxide, high hydrostatic pressure, pulsed electric fields, short-wave UV, ultrasound, and gamma radiation are being applied and/or studied [8,9,10,11,12,13,14]. However, the high sanitary requirements of foods and the consumers’ concern regarding the degree of intervention involved in their processing are generating a need to implement new sanitation techniques with low impact on products [15]. 

Nonionizing electromagnetic fields (EMF) have been used for the control of foodborne pathogens such as *Escherichia coli*, *Listeria monocytogenes,* and *Staphylococcus aureus*. It has been found that the use of 0.1 to 50 Hz or 18 to 73 GHz frequencies has reduced the viability of the microorganisms by damaging their cell membranes [16,17,18,19,20,21]. Akbal and Balik [22] studied the effect of the exposure of *E. coli* and *Bacillus subtilis* to nonionizing EMF generated by a mobile phone, and found that a constant 10-h exposure to 900 MHz affects the growth curve of *E. coli*, reducing its optical density in a nutritive liquid media. On the other hand, Latif et al. [23] showed that there are changes in the genome of *E. coli* and *S. aureus* caused by exposure to a 900 MHz EMF. Rodriguez et al. [15] studied the effect of EMF on the viability of *E. coli* ATCC25922 in peptone water. After a 48-h exposure to a 2.412 GHz and 2.462 GHz, it was determined that there is statistically significant inhibition of the bacterial growth, with the inhibition value being greater than one logarithmic cycle. These studies showed the antibacterial capacity presented by EMF, but there are fewer studies on yeasts. Therefore, the objective of the present work was to analyze the effect of the EMF between 1 to 6 GHz on the growth of *Saccharomyces cerevisiae* and to establish a possible mechanism of antifungal action.

## 2. Materials and Methods

### 2.1. Yeast Growth Conditions

For this study, the commercial *S. cerevisiae* strain LALVIN EC-1118 was used (Lallemand Inc., Montreal, QC, Canada), which was obtained from the Culture Collection of the Biotechnology and Microbiology Laboratory at Universidad de Santiago de Chile.

### 2.2. Exposure of S. cerevisiae to Several EMF Frequencies at Different Distances

The electromagnetic chamber was a 1-mm thick stainless steel 50 × 50 × 70 cm container equipped with three antennas that received the signals from a BPSG6 generator (Aaronia, Germany) with a frequency range from 23.5 MHz to 6 GHz, a precision of 15 Hz, and an output power from −45 to 18 dBm. The antennas were arranged mutually orthogonally, allowing a high electromagnetic density inside the chamber, and a platform was installed, allowing exposure at distances from 2 to 30 cm from the radiation center (Figure 1).

For the exposure, *S. cerevisiae* was inoculated in 5 mL of YPD medium (yeast extract 5 g/L, peptone 5 g/L, and glucose 20 g/L) and incubated at 28 °C in an orbital agitator at 120 rpm, until a density of 1 × 10^6^ cells/mL. Then, 10 µL were placed in the center of a pair of sterile square overlapping 2 × 2 cm glass plates (experimentation model called “sandwich”). Each sandwich was placed independently in plastic Petri dishes, and it was then subjected to the experimental conditions in the electromagnetic chamber (Figure 1). The Petri dishes were placed at distances of 2, 4, 6, 8, 10, and 25 cm from the antennas emitting the waves. The exposure to the EMF was for 15 min at frequencies of 1.0, 3.0, and 5.9 GHz at 18 dBm. As the control, the same tests were repeated in the absence of EMF, using a chamber of the same material and geometry. All the exposure conditions were made in triplicate at a controlled temperature of 25 °C.

### 2.3. Determination of the Percent Viability Reduction of S. cerevisiae after Irradiation

Immediately after the exposure of the “sandwich experimental models” to the EMF, the glasses were separated, and 30 mL of peptone water 25 g/L (Merck, Boston, MA, USA) was added in the same Petri dish. To facilitate the resuspension of the yeasts, the Petri dishes were shaken orbitally at 70 rpm for 15 min at room temperature. From each plate 10 µL of the suspension were seeded on YPD plates (supplemented with 20 g/L of agar) and they were incubated for 48 h at 28 °C. After counting the colony forming units (CFU), the percentage of cell viability reduction was determined using the calculation established in Standard Test Method ASTM E2149-10 [24]. A similar protocol was followed for the control sample.

### 2.4. Exploration of the Antifungal Action Mechanism of the EMF on S. cerevisiae

*S. cerevisiae* at 3 × 10^5^ cells were exposed individually to EMF under the conditions identified in this work (see Section 3.2). This was carried out three times to assess the possible effects:

(a) Cell wall: to observe the cells location in fluorescents fields, *S. cerevisiae* was stained with calcofluor white (CW), which joins the chitin present in this structure [25]. The cell wall was stained with CW (Sigma-Aldrich^®^, St. Louis, MO, USA) in a 1:1 proportion with KOH at 10%. (b) Membrane permeability: the membrane damage was determined indirectly using propidium iodide (PI), which stains nucleic acids of yeasts which have lost their impermeability [26]. Then the cells were stained with 2 µM propidium iodide (Sigma^®^) (c) Reactive oxygen species (ROS): to determine if this fraction is related to the production of ROS, the cells were stained with 10 µM 6-carboxy-2’,7’-dichlorodihydrofluorescein diacetate (C400, Thermo-Scientific^®^, Waltham, MA, USA). This redox stain reacts with ROS in the cell cytoplasm, producing the accumulation of its reduced form [27]. As the positive control, a similar number of cells were treated with 600 µg/mL of zymolyase 100*T* (Amsbio^®^, Abingdon, UK) during 2 h at 37 °C and then exposed to 30% *v*/*v* H_2_O_2_ during 30 min at room temperature. As the negative control, the same concentration of cells was incubated first 2 h at 37 °C and posteriorly for 30 min at room temperature in HEPES saline 1× pH 7.0 buffer (70 mM NaCl, 0.75 mM Na_2_HPO_4_, 25 mM HEPES), similar to the positive control. After the treatments, the cells were washed 3 times with HEPES saline 1× pH 7.0 buffer. The fluorescent cells were observed using the epifluorescence microscope Moticam Pro BA410 (Motic^®^, Hongkong, China), with 40× fluorescence microscope objective.

### 2.5. Observation by Transmission Electron Microscopy (TEM)

*S. cerevisiae* at 3 × 10^5^ cell/mL were exposed to EMF under the conditions established in this study. The cell suspension was then centrifuged for 10 min at 5900× *g* and the pellet was fixed in 2.5% glutaraldehyde in 0.1 M pH 7.0 sodium cacodilate buffer during 6 h at room temperature, and then washed in cacodilate buffer overnight. This was post-fixed in 1% aqueous osmium tetroxide for 90 min, then washed for 30 min with distilled water, and they were block-stained with 1% aqueous uranyl acetate during 60 min. It was dehydrated in a battery of 50%, 70%, 95%, and 100% acetone for 20 min each. It was pre-included in 1/1 epon/acetone overnight and then it was left in pure epon for 4 h at ambient temperature. It was included in pure resin in an Eppendorf tube and polymerized in an oven at 6 °C during 48 h. Fine slices (80 nm) were obtained in a Leica Ultracut R ultramicrotome and they were received on a 300-mesh copper grid. The slices were stained with 4% uranyl acetate in methanol for 2 min and in lead citrate during 5 min, and they were examined in a Philips Tecnai 12 Biotwin (Maastricht, The Netherlands) transmission electron microscope at 80 kV. This analysis was done in Advanced Microscopy Unit, Universidad Católica de Chile.

### 2.6. Statistical Analysis

The data distribution analysis was made using the Kolmogorov–Smirnov test. The parametric data were analyzed according to their statistical variance (ANOVA), while the non-parametric data were analyzed using the Kruskal–Wallis test. The significant differences were validated with a probability *p* < 0.05.

## 3. Results

### 3.1. Study of the Effect of EMF on the Viability of S. cerevisiae

This work studied the effect of the exposure to EMF on the growth of *S. cerevisiae*. The yeast was exposed to three EMF bands (1.0, 3.0, and 5.9 GHz) at different distances from the radiation center (point 2.2), and was then seeded on the culture plates, thus determining the cell viability (Figure 2). This figure shows that exposure of *S. cerevisiae* to 1 GHz causes a reduction of the viability at all the exposure distances. However, at distances of 10 cm and 25 cm from the radiation center, the largest reductions of the yeast population were achieved, reaching 58.9% and 33.5%, respectively. On the other hand, when the cells were exposed to 3 GHz, the reduction varied between approximately 5% and 25%, with the exposure at 4 cm from the radiation center having the largest effect, with a 25.1% decrease of the yeast population. At a frequency of 5.9 GHz, there was a greater variation of the cell viability, with 18.8% reduction at a distance of 8 cm from the radiation center. The results show that the EMF in the studied band affects the viability of the *S. cerevisiae* yeast, as well as showing that there is a relation between the exposure distance and the frequency used.

### 3.2. Effect of the EMF on the Cells

Cell suspensions were submitted to 5.9 GHz at distances of 2, 6, and 8 cm from the center of the radiofrequency emission. They were then treated with different dyes to determine a possible action mechanism. In column CW, Figure 3A,D,G,J,M show the cells stained with CW, determining the number of yeast cells present in the field for each treatment. On the other hand, in column PI, Figure 3B,E,H,K,N show the cells stained with PI to determine the membrane’s permeability; in column C400, Figure 3C,F,I,L,O show the staining with C400, allowing determination the production of intracellular ROS. In the case of the controls: (a) negative control: cells untreated with EMF showed that there were no cells stained red with PI, or green with C400. On the other hand, (b) positive control: cells were treated with zymolyase and H_2_O_2_ caused a significant increase of cells stained red and green (Figure 3E,F). Then, both controls validated the experimental method used in this study.

With respect to the effect of the treatment to which the cells were submitted, Figure 3H,K,N show a sustained increase of yeast cells with red fluorescence, indicating that these cells have lost their impermeability to PI, which means that in some way the cell membrane has been damaged. On the other hand, Figure 3I,L,O show a slight increase of the green florescence as the distance from the emission center of the stimulus is increased, indicating that such a frequency would not generate an increase of the ROS in the yeast. 

The statistical comparison of each treatment in relation to the exposure distance (Figure 4) shows that there is an increase of the proportion of cells stained red with PI as the distance from the frequency emitting center increases, which is statistically greater than that of the yeasts without treatment (negative control) and statistically similar to the membrane damage produced by the treatment with zymolyase and H_2_O_2_ (positive control). 

On the other hand, by evaluating the appearance of intracellular ROS, it can be seen that there is a statistically significant increase of green yeasts exposed to 5.9 GHz at 2 and 8 cm from the frequency emitting center compared to the yeasts not exposed to this stimulus (negative control). However, this increase is less compared to the positive control, where the yeast was subjected to a treatment that induced the accumulation of intracellular ROS (positive control). 

Based on these results, it is shown that the exposure of *S. cerevisiae* to a 5.9 GHz EMF would cause a loss of the impermeability of the cell membrane. To visualize this fact, a TEM of the cells exposed to the 5.9 GHz frequency and unexposed cells to EMF (negative control) was carried out. In Figure 5 the untreated yeast cells have a whole cell membrane, which appears as a continuous cord that separates the cell cytoplasm from the wall and the external environment (Figure 5A, white arrows). In the case of the yeast exposed to EMF, zones where the continuity of the membrane is lost are seen, without the continuous presence of this cord (Figure 5, black arrows), in contrast with zones where the structure is still complete (Figure 5B, white arrows). 

## 4. Discussion

EMF has been widely used in the food industry and the most common is the 2.450 GHz band used in microwave ovens. 13.56, 27.12, and 40.86 MHz bands have been used commercially in the dehydratation of textiles, paper, and some food products [28]. In the microbiological field, the use of UV radiation, which belongs to the (100 to 200 nm) ionizing band, has been studied extensively [29,30]. Also, there are studies on other frequency bands, such as nonionizing (1–300 GHz) for the control of microorganisms. Shamis et al. [31] showed a decrease of the *E. coli* population when this microorganism was exposed to 18 GHz. Furthermore, Akbal and Balik [22] exposed *E. coli* and *Bacillus subtilis* cultures to 1800 MHz from 1 to 10 h and observed a decrease of the growth of *E. coli* compared to the control test (without exposure), In the case of *B. subtilis*, no effects were observed. Mulye et al. [32] exposed *E. coli* cultures to mobile telephone towers emitting (95–57 dBm) radiation, finding a decrease in the viability of the bacteria. At a frequency of 2.45 GHz, there may be an alteration of the structural and functional properties of the cell membrane, such as ion transport and permeability to macromolecules [33]. In the present work, it was found a negative effect on the growth of *S. cerevisiae* when it is exposed to the 1 to 5.9 GHz EMF band. Furthermore, it was seen that this effect would depend on the frequency used, as well as on the distance from which it is emitted. A frequency of 1.0 GHz had the most negative effect on the microbial growth, reaching 58.9% growth at a distance of 10 cm from the center of the radiation emission, followed by 33.5% viability at that same frequency, but at 4 cm from the center of the emission. On the other hand, at a frequency of 5.9 GHz a cycle form was found, achieving the largest decrease of the yeast population at 4 and 10 cm from the emission center, with 5% and 10%, respectively. Furthermore, in the case of the intermediate frequency used in this study, 3 GHz, 4 cm from the emission center produced the largest effect achieved at this frequency, reaching a decrease of 25.1%. This may indicate that lower frequency bands require a greater distance from the emission center of the stimulus to exert their effect on the microorganism, and that, in contrast, increasing the value of the band would require shorter distances. Grundler et al. [34] showed that the influence of this stimulus, either positive or negative on the growth of *S. cerevisiae*, depends on the frequency of the band to which the yeasts are exposed, but with no direct relation with the band frequency. Nguyen et al. [19], exposing different microorganisms (*Branhamella catarrhalis*, *Kocuria rosea*, *S. aureus*, *Streptomyces griseus*, and *S. cerevisiae*) to 18 GHz, found an effect on the permeability of the cell membrane. Through a study with TEM, they determined that the irradiation allowed the entrance of neutrally charged silica nanospheres 23.5 ± 0.2 nm into the cells. On the other hand, Dreyfuss and Chipley [35] showed that in *S. aureus*, the microwave radiation affects the metabolic activity which cannot be explained by thermal effects alone. Similar observations were made by Al-Harbi et al. [16] using a very low frequency EMF of 0.3 Hz. In that paper, the authors reported that by TEM, the exposure of *E. coli* for 90 min to that EMF caused irreversible damage to the cell membrane and the complete disorganization of its content during the binary fission processes. Shamis et al. [31] showed that, in the case of *E. coli* when it is exposed to 18 GHz, pores would appear on the cell membrane, affecting the growth of this microorganism. These authors propose that microwave charging and discharging controls electrical behavior and molecular transport. This background would agree with the observations presented in our work, where exposure during 15 min to an EMF of 5.9 GHz caused the permeabilization of *S. cerevisiae* cell membrane. In this context, Novickij et al. [36] studied the effect of conventional electroporation combined with the application of electromagnetic pulses on the membrane permeabilization capacity in the pathogenic yeast *Candida albicans*. The result was that the exposure of *C. albicans* to a treatment of 50 electromagnetic pulses (3.3 T each) added to an electric pulse of 17 kV/cm made approximately 80% of the yeast cells lose their membrane permeability (measured through the penetration of PI into the cell). Additionally, the authors describe this treatment that causes a 21% decrease of the viability of *C. albicans*, clearly showing the antifungal effect of exposure to this EMF [36]. However, it has been reported that the growth rate of the *S. cerevisiae* yeast increases by up to 15% or decreases by 29% when microwave frequencies between 41.8 and 42.0 GHz are used [34]. Similar observations were made when Zeng et al. [33] treated the *Brettanomyces curtensii* yeast with different exposures to microwave irradiation. The same authors mention that cell death may be due to an irreversible increase of electrolyte and Ca^2+^. This background reinforces the observations obtained in our work by means of fluorescence microscopy and TEM, where it was seen that when *S. cerevisiae* is exposed to 5.9 GHz and 18 dBm for 15 min there would be irreversible cell damage that would be responsible for the death of the yeast.

## 5. Conclusions

This work presents evidence that allows concluding that exposure of *S. cerevisiae* to nonionizing 1.0, 3.0, and 5.9 GHz EMF at different distances from the radiation center causes decreased viability of the microorganism that was studied. Exposure to 1.0 GHz band caused the greatest reduction at distances greater than 8 cm. Exposure of the yeasts to 3.0 and 5.9 GHz caused an oscillation of the viability depending on the exposure distances, giving the maximum reduction values at 4 and 8 cm, respectively. Comparing the percentages obtained at the different distances between the emission center and the sample it was found that at 6 cm, the three bands caused decreased viability. This can be explained because the distance covered would cause a decrease of the energy of each frequency. The EMF would produce a damage at the cell membrane level of the yeasts, without substantial increases of the intracellular ROS levels. The results obtained in this study support the use of EMF to be applied in the food industry as a method for the control of the growth of microorganisms.

## Figures and Tables

**Figure 1 jof-07-00281-f001:**
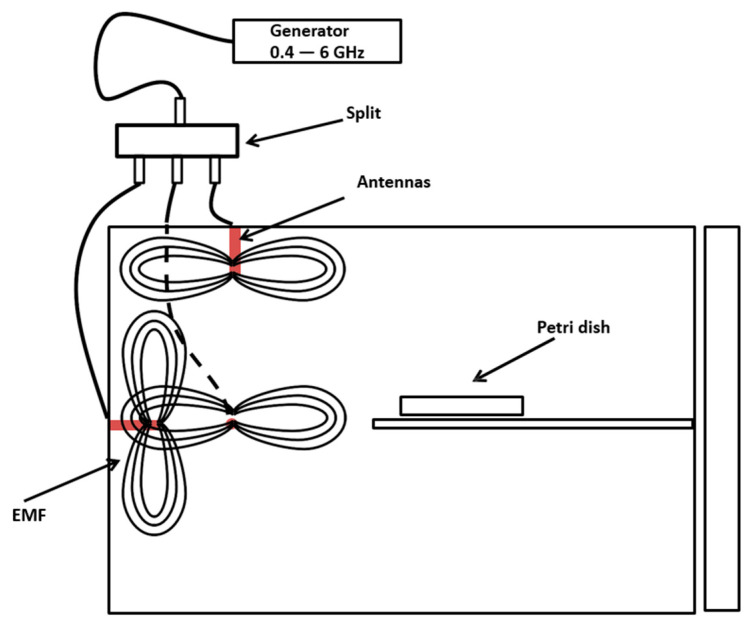
Schematic representation of the electromagnetic chamber.

**Figure 2 jof-07-00281-f002:**
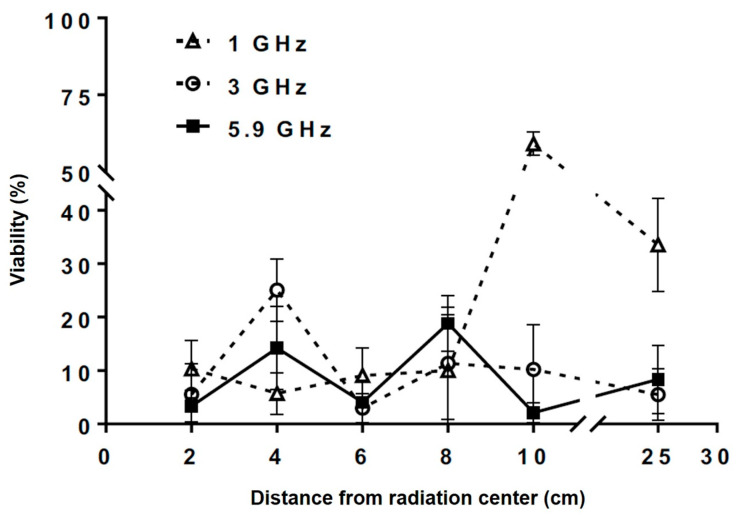
*S. cerevisiae* viability after 15 min of exposure to 1 GHz (striped line/white triangles), 3 GHz (striped line/white circles), and 5.9 GHz (black line/black squares) at different distances from the antennas. The gaps in both axes represent changes in the scales.

**Figure 3 jof-07-00281-f003:**
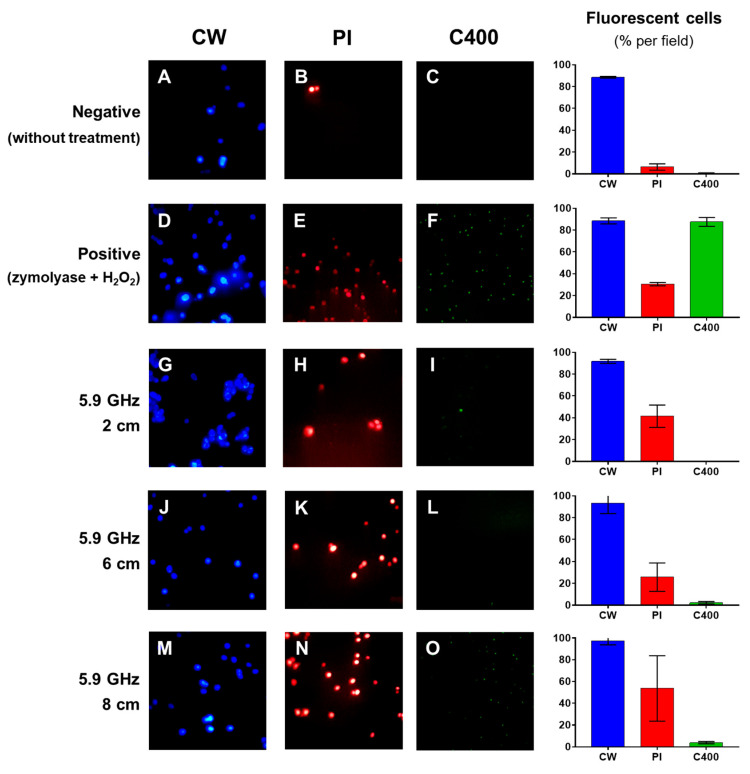
Fluorescence microscopy of *S. cerevisiae* EC1118 exposed to 5.9 GHz for 15 min at different distances from the antennas. The stains used were calcofluor white (CW; blue), propidium iodide (PI; red), and 6-carboxy-2’,7’-dichlorodihydrofluorescein diacetate (C400; green). The graphs on the right of each line represent a percentage of fluorescent cells per field counted in each treatment in duplicate. (**A**–**C**): untreated yeasts (negative control). (**D**–**F**): yeasts after treatment with zymolyase (600 µg/mL) before exposure to H_2_O_2_ 30% *v*/*v* for 30 min (positive control). (**G**–**I**): yeasts exposed at 2 cm from the antenna. (**J**–**L**): yeasts exposed at 6 cm from the antenna. (**M**–**O**): yeasts exposed at 8 cm from the antenna.

**Figure 4 jof-07-00281-f004:**
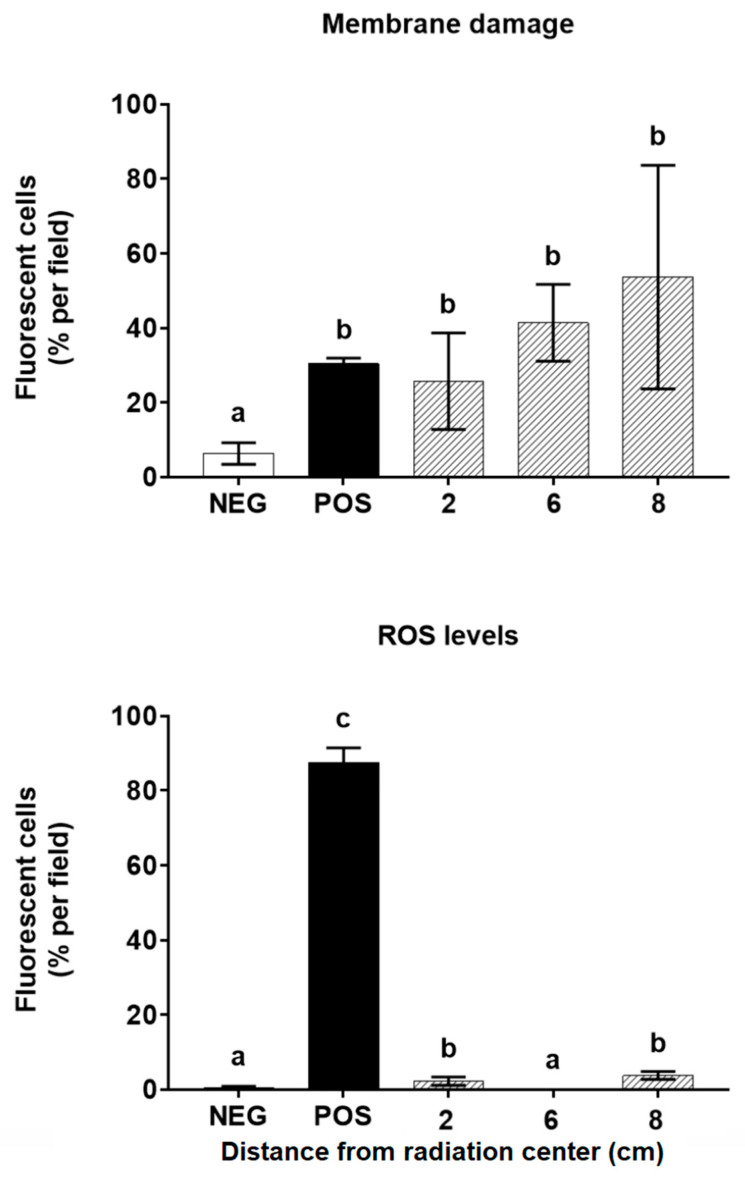
Evaluation of cell damage in *S. cerevisiae* EC1118 after exposure for 15 min to a 5.9 GHz band at different distances from the antennas. The membrane damage and ROS levels were measured using the percentage of fluorescent yeasts obtained after staining with propidium iodide (PI) and 6-carboxy-2’,7’-dichlorodihydrofluorescein diacetate (C400), respectively. White bars (labeled NEG) correspond to untreated cells (negative control), while black bars (labeled POS) correspond to yeasts treated with zymolyase (600 µg/mL) before exposure to H_2_O_2_ 30% *v*/*v* for 30 min (positive control). Bars in striped lines correspond to exposure distances (cm) from the antennas. Different letters above each bar represent a statistical difference between the treatments (*p* < 0.05).

**Figure 5 jof-07-00281-f005:**
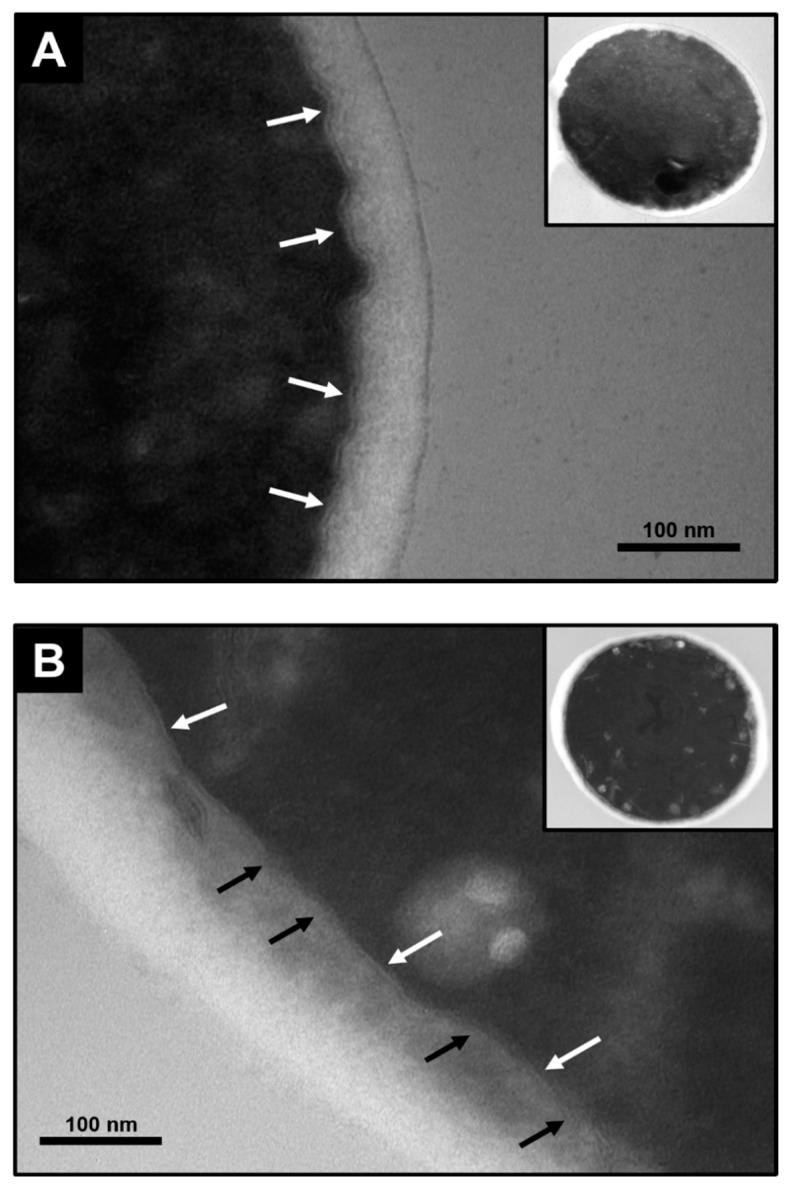
Cell membrane damage in *S. cerevisiae* EC1118 after exposure for 15 min to a 5.9 GHz at 8 cm from the antennas, using Transmission Electron Microscopy (TEM). (**A**) yeast without treatment. (**B**) yeast treated. White arrows show the membrane integrity (seen as a regular cord surrounding the yeast cytoplasm), while black arrows show regions with membrane damage (seen as loss of membrane continuity). The membrane was observed at a magnification of 135,000 times the regular cell size (~5 µm).

## Data Availability

Not applicable.

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
