# Peer review of "Nonionizing Electromagnetic Field: A Promising Alternative for Growing Control Yeast"

_jof, 2021, doi:10.3390/jof7040281_

Round 1

Reviewer 1 Report

This article presents an interesting possibility of using EMF to control yeast growth for products preservation. This work use a single yeast strain, S cerevisiae LALVIN EC1118 so it would be very interesting to check if other species or strains show similar results.

Regarding materials and methods S. cerevisiea straisn is grown at 28C, its optimal growing temperture , I suppose, but the experiment of exposure to different frequencies was carried out at 25C Why this difference?

Similary, in the experiment of explotarion of the antifungical mechanism of the EMF, the ROS determination is made at room temperature, the negative control, of this experiment, at 28C and the possitive control at 37C .

In the Figure 4, a, b, c letters represent sigificative difference p>0.05, so what is de difference between the different letters?

Minor considerations

Figure 3. In negative control image F, green fluoresence is not observed due to the quality or edition of the photograph, I suppose.

line 279 the is work/ the are studies on other....

Author Response

We made a careful revision of our manuscript in the light of your comments.

Comments

Q1: Regarding materials and methods S. cerevisiea strains is grown at 28C, its optimal growing temperature, I suppose, but the experiment of exposure to different frequencies was carried out at 25C Why this difference?

R1: In our experimental procedure, we used 28 °C as an optimal growth temperature to cultivate the strain LALVIN EC1118 in the laboratory medium. The exposition of yeast cells to electromagnetic fields was made inside of a stainless-steel chamber. However, the electromagnetic chamber lack of temperature control system. However, we placed the chamber in a temperature-controlled room, pre-set at environmental referential conditions (25 °C, 1 ATM), established by the American Environmental Protection Agency (EPA). This information was added to the manuscript in lines 107-108 (highlighted in yellow).

Q2: Similarly, in the experiment of exploration of the antifungal mechanism of the EMF, the ROS determination is made at room temperature, the negative control, of this experiment, at 28ºC and the positive control at 37ºC.

R2: We agree with your comment. We detected a writing mistake in the manuscript (Materials and Method, point 2.4). In our experimental procedure, we incubated the yeast cell corresponding to positive control at 37 °C for 30 min, because it is the optimal temperature to zymolyase activity. After that, we worked at room temperature. In the case of negative control, the cells were exposed at 37 °C for 30 minutes in buffer HEPES saline 1X before the stain procedure, mimicking the treatment made in the positive control. The corrections applied in the manuscript were highlighted (with yellow) in lines 133 – 139, 225 – 226 and 252 – 253.

Q3: In the Figure 4, a, b, c letters represent significative difference p>0.05, so what is de difference between the different letters?

R3: Different letters above each column represent a statistical difference between the bars. From another point of view, the treatments represented with bars labeled with letter “a” are statistically different with the treatments represented in bars labeled with other letter (b or c). In order to make a clear explanation, we modified the last phase on Figure 4 (lines 254 – 555 highlighted in yellow).

Q4: Minor considerations

Figure 3. In negative control image F, green fluorescence is not observed due to the quality or edition of the photograph, I suppose.

R4: In Figure 3 we showed the number of cells which present fluorescence derived from each treatment, and subsequently, we studied the percentage of fluorescent cells, independent of their   intensity. As we explained in lines 134-135 (highlighted in yellow), the fluorescence of C400 was dependent on the amount of ROS contained in the cell cytoplasm. Therefore, the intensity of color trends was weak, affecting the observation of fluorescent cells showed in the letter F (Figure 3).

Q5: line 279 the is work/ the are studies on other....

R5: We agree with your comment. We corrected this phrase on the manuscript (line 281, highlighted in yellow)

Finally, we have realized a careful linguistic revision of the whole manuscript.

We kindly thank you for your careful review of our manuscript.  

Reviewer 2 Report

The manuscript presents original results. It is significant in content. The article will be in high interest to the readers. The results are very useful for the practice from the technological point of view. The abstract is concise. The introduction provides sufficient background of the research and describes clear hypothesis of the present study. The research design is appropriate and the methods used are comprehensively described and cited. The appropriate statistical tests are used and the results are correctly reported. Included figures are clear and accurately present the results. The results are discussed from the view of currently available knowledge in the research field. The available data are analyzed and interpreted and critical reviewed. The references used are sufficient in number and cover the available information on the topic. Conclusion highlights the main aspects on the study. The language is clear and understandable.

Author Response

Thank for your comments.